# Global Food Security and Sustainability Issues: The Road to 2030 from Nutrition and Sustainable Healthy Diets to Food Systems Change

**DOI:** 10.3390/foods13020306

**Published:** 2024-01-18

**Authors:** Theodoros Varzakas, Slim Smaoui

**Affiliations:** 1Department of Food Science and Technology, University of the Peloponnese, Antikalamos, 24100 Kalamata, Greece; 2Laboratory of Microbial, Enzymatic Biotechnology, and Biomolecules (LBMEB), Center of Biotechnology of Sfax, University of Sfax-Tunisia, Sfax 3029, Tunisia; slim.smaoui@cbs.rnrt.tn

**Keywords:** governance, food safety, food system transformation, sustainability transitions

## Abstract

The accomplishment of food/nutrition security for all across sustainable food systems (SFS) is tied to the Sustainable Development Goals (SDGs). SFS is connected to all SDGs via the traditional framework of social inclusion, economic development, environmental safety, inclusivity, and the development of sustainable food systems. We suggest that, for the world to achieve sustainable development, a shift to SFS is necessary to guarantee food/nutrition security for all, while operating within planetary boundaries to protect ecosystems and adapt to and mitigate climate change. Therefore, there is a requirement for original approaches that implement systemic and more participatory methods to engage with a wider range of food system stakeholders. However, the lack of skills and tools regarding novel methodologies for food system transformation is a key obstacle to the deployment of such approaches in practice. In the first part of this review, a summary of some challenges that occur in the governance of food system transformation is given. Through a case study of plant-based proteins and their biological and chemical modification as diets shift towards alternative proteins, we demonstrate that resource-efficient food systems and food waste, through system transformation, are useful in understanding both (i) how food system transformation has ensued and (ii) how the required transformation is prohibited. Finally, we discuss the implications of food system transformation in terms of nutrition and sustainable healthy diets, which are needed to achieve changes in food safety systems in the future. The linkage of food and the environment is evident, focusing on nutrition and sustainable healthy diets. This cannot be accomplished without system change and research towards new foods and, more specifically, new proteins such as plant-based ones and their biological and chemical modification.

## 1. Introduction

Owing to the pressures provoked by the present, allied, global food systems leading to health/environmental degradation, challenges to redevelop them to be more sustainable are progressively emerging across the world. Above all, the tendency is to change from individualized agendas to cooperative strategies that can successfully promote the authentic transformation of food systems to be more sustainable. In this sense, according to the European Commission (2020) [1], food system transformation is required in order to shift towards a more sustainable and healthy diet, ensuring holistic food and nutrition security. Hence, a more thorough and comprehensive understanding of the different components of the current food systems and their interactions is required for the maximum co-benefits.

Future-proof sustainable food systems with a focus on health and inclusion are a key focus of the European Commission (EC). The Farm to Fork strategy and the European Green Deal policy are important tools related to the success of these food systems. Food 2030, an EU research and innovation policy framework, supports the transition towards maintainable, innovative, and comprehensive food systems that respect planetary boundaries. Human health, the climate, the planet, and communities will all benefit from the implementation of these systems.

The United Nations Sustainable Development Goals (UN SDGs) [2] and the European Green Deal [3] are considered essential to mitigate the anthropogenic climate change (CC) crisis. They are synergetic since they endorse maintainable agrifood systems and the preservation of the environment [4].

The Sustainable Development Goals (SDGs), approved by the UN in 2015, include several worldwide goals focused on accomplishing a maintainable future for all by 2030 [5]. It should be noted that the change in food systems is crucial due to their unsustainable nature. Climate change, resource scarcity, effluence and waste, environmental degradation, biodiversity damage, human development, undernourishment, and diet-connected non-communicable diseases are all drivers heavily affecting this change.

Food 2030 requires that the entire food system is linked together, associating multiple sectors from farm to fork, i.e., from primary production and food processing to retailing and distribution, food services, and consumption.

All stakeholders should be involved in this process, engaging science–policy–society (consumers). In this way, research and innovation policy will be improved, aiming at coherence and stability, and research and innovation funding and investment will be increased. Hence, the consumer should be an integral part of this process. Finally, the role of innovative technologies should be supported and initiated, along with new approaches and business models, accompanied by social, institutional, and governance innovation relevant to food system change (Figure 1).

The co-benefits of these four thematic priorities are presented in the following.

### 1.1. Nutrition for Maintainable and Healthy Diets

It is important to tackle important issues such as malnutrition and obesity, the support of healthy ageing, the development of novel protein substitutes towards plant-based diets, the improvement of food authenticity/traceability, the support of the cultivation and consumption of overlooked harvests for nutrition and resilience, and the support of the shift towards sustainable healthy diets in Europe and Africa.

Additional expansions and applications of EU food directives and food safety policies (Food Safety—European Commission (europa.eu)) and the Expert Group on Public Health—European Commission (europa.eu) (https://ec.europa.eu/health/non_communicable_diseases/steeringgroup_promotionprevention_en and https://ec.europa.eu/jrc/en/ health-knowledge-gateway) (accessed on 28 December 2023) are relevant to Sustainable Development Goals 2, 3, 8, and 10 [1].

### 1.2. Food Systems Supporting a Healthy Planet

Water, soil, land, and sea should be managed dependably, thus making them available in the future. Smarter food systems are the priority of Food 2030. Hence, they will be better aligned with climate change, and, in this way, will help to preserve the environment. In this direction, environmental risks will be limited and the flow of greenhouse gases into the atmosphere will be reduced [1]. The priority is to devise and operate environmentally friendly and resilient food systems that boost biodiversity, fostering sustainable and healthy agriculture and aquaculture.

### 1.3. Circularity and Resource Efficiency

The aim of circularity and resource efficiency is to use more efficient and greener industrial processes and logistics in order to reduce food, water, and energy waste. This can be achieved by using unavoidable biomass and waste resources. Another solution is the provision of local food on demand for short supply chains.

Circularity can be defined by the use of maintainable, resource-effective food systems that can manage the 1.3 billion tons of food lost and wasted each year. This could be achieved by zero food waste (FW) policies, the efficient recycling of food waste, the biodegradation of food packaging, limiting microplastics, and responding to the increased demand for more local and healthy food [1].

### 1.4. Innovation and Empowering Communities

The development of an ecosystem supporting new business models and solutions for society is the aim of the fourth Food 2030 priority.

The achievement of this goal will help to link urban/rural/coastal economies and establish communities across the EU. Closer linkages and partnerships among industry and society will help to create new jobs, decrease prices, and enhance sustainability. Key challenges in this direction include governance innovation, social innovation through citizens’ involvement, citizens’ engagement in food science and policy, a farm to fork economy with a focus on social innovation, and the development of data-driven food and nutrition systems with the goal of meeting societal needs [1].

### 1.5. Impact of Climate Change on Food Security

Biodiversity is an essential source of food. In this context, an awareness of species disappearance is necessary, caused by factors such as pollution, pests, and food and medicine control. As an illustration, between 1996 and 2003, the precipitation in parts of equatorial East Africa provoked flooding and reductions in crops and agricultural yields [6]. Consequently, climate change has a direct impact on food production and distribution [7]. Firstly, an increase in the incidence of pests and diseases has been observed, and a loss of biodiversity and a decline in ecosystem functioning has been noted. Secondly, the accessibility of water for crops and fish production and a sea-level rise has been observed [8]. The impacts include the loss of life and food security of millions of people in disaster-prone areas. Through extreme weather, CC will disturb food security and crop yields too. By 2050, it is projected that agricultural yields in Africa alone could decline by >30% [9].

On the other hand, food preparation, processing, acquisition, distribution, and consumption are impacted by CC [10], which influences plant and animal growth, water cycles, biodiversity and nutrient cycling, and the ways in which these are managed for agricultural practices and food production [11]. In addition, CC could amend suitable cultivation zones with a wide range of crops.

CC also influences on income-earning balances, which could affect the ability to buy food, and a changing climate or climate extremes may affect the availability of certain food products. For example, in Tunisia and Egypt, there have been augmented prices for basic foodstuffs [12].

CC has augmented the genetic erosion of landraces and threatens wild species, including crops’ wild relatives [13]. As a result, the existing varieties could be lost as farmers replace them with other landraces and improved varieties that are better adapted to the new conditions.

In this review, a summary of some challenges that occur in the governance of food system transformation is firstly given. Through a case study of plant-based proteins and their biological and chemical modification due to the dietary shift towards alternative proteins, we demonstrate that resource-efficient food systems and FW on system transformation is useful in understanding both (i) how food system transformation has ensued and (ii) how the required transformation is prohibited. Finally, we discuss the implications for governing food system transformations in terms of the nutrition and sustainable healthy diets that are needed to implement changes in the food safety systems of the future (Figure 2).

## 2. Governance and System Change

The first pathway for action is governance and system change. Food systems need to be resistant towards global challenges such as CC, but this will depend upon the successful development, integration, and implementation of policies and strategies, such as the Green Deal and Farm to Fork strategy. The latter will set out an R&I policy and this will lead to effective food system transformation [1].

In this context, the protection and restoration of natural ecosystems, along with the sustainable use of resources, can lead to improvements in human health. Benefits in this direction include climate impartiality and adaption to climate variation. Biodiversity increases and natural resource exploitation will be useful in order to apply farm to fork food policies and nutrition security. In order to achieve these benefits, the consideration of the interactive, socioeconomic/demographic drivers of change is necessary for the redesign of sustainable rural, coastal, peri-urban, and urban areas. It is noteworthy that zero pollution effects should be considered, as noted in the 11th pathway, as well as a fair and democratic environment [1].

An example of this is the ‘European SUStainable Food and Nutrition Security’ (SUSFANS) project (2015–2019), with the objective of building a framework, evidence base, and analytical tools for EU-wide food policies and their association with consumer diets. Finally, their implications for nutrition and public health in the EU have been outlined in SUSFANS PUBLICATIONS|SUSFANS [14].

According to Rutten et al. [15], there should be a correlation between improvements in European diets and sustainable food systems. Metrics, models, and navigation tools regarding sustainable food and nutrition security have been employed.

SUSFANS developed metrics, identified and analyzed drivers, integrated data and modeling, and formulated foresight for EU sustainable food and nutrition security (FNS). It developed an analytical toolbox integrating new and improved micro-level models of nutrient intake with habitual dietary patterns and the preferences of consumers, along with health impacts [16,17]. Connections among the agri-food/fish sectors with the energy sector, factor markets, labor supply, and health sector and intercontinental employment were assessed by the Computable General Equilibrium (CGE) model MAGNET. However, partial equilibrium (PE) economic and biophysical models in the forest/crop/livestock sectors were analyzed by the Global Biosphere Management Model (GLOBIOM). In the latter, the CC impacts on global agriculture and food availability were discussed thoroughly. In addition, EPIC, the biophysical crop growth model, provided management system-specific weather and soil information.

Food production seems to be affected by CC and plant variability effects. The physiology of crops seems to be affected by the gradual increase in temperatures throughout many regions, hence leading to production and quality limitations [18] Moreover, the capacity of food production is reduced, leading to major economic losses derived from unpredictable cycles of drought and excess humidity in crops [19,20,21,22]. This might favor the appearance of new variants of pests and diseases, which will be uncontrollable and aggressive [23].

Jimenez et al. [24] exploited the possibility of using native microbiota as a practical alternative to converse pliability on harvests. Anthropogenic emissions of greenhouse gases (GHGs) affect the development and lifestyles of society, leading to CC [25]. These effects cause irreversible impacts [26]. The Intergovernmental Panel on Climate Change (IPCC), in its sixth assessment report (AR6) in 2018, estimated that, since pre-industrial times (1850–1900), global warming of approximately 1.1 °C was caused by human activities by 2021 [27].

A good solution to the food crisis in many areas of the world is represented by changes in agricultural, forestry, or livestock processes. During 2007–2016, these activities represented around 13% of global carbon dioxide (CO_2_) emissions, 44% of methane (CH_4_), and 82% of nitrous oxide (N_2_O) [28]. Food security is heavily affected by extreme temperatures since many crops are destroyed. Hence, areas of cultivation are being reduced, along with effects on livestock in tropical areas.

The three most significant aspects for changes in precipitation in the future account for (a) an increase in precipitation intensity [29]; (b) hurricanes and tropical storms, causing an increase in the frequency and speed of winds [30]; (c) changes in the start and end of the rainy season in tropical areas [27]. These implications will undoubtedly affect crop cycles, agricultural production, and later post-harvest stages, hence affecting food systems.

In order to reduce GHG emissions, bio-based products should be manufactured using eco-efficient and robust technologies for the processing of biomass and waste in biorefineries [31]. These bio-based production methods and new processes should generate new biomolecules for biostimulation, biocontrol, and fertilization (e.g., struvite, integrated biochar–compost), leading to innovation in crop protection [32]. The bioeconomy and agroecology could also constitute a synergistic solution for the further development of these new climate-smart agricultural systems, as reported by Faucon et al. [33]. They reported on the ecological role of crop diversification, waste recycling, and biomass transformation for agroecological development. Finally, they discussed the holistic approach of the combination of agroecology and the bioeconomy for sustainable agricultural systems.

## 3. Urban Food System Transformation

In order to understand urban food system transformation, it is essential to consider science-based multi-actor governance processes [1].

Nowadays, urban areas accommodate >50% of the world’s populace [34], with an estimated increase of over 70% by 2050 [35,36]. Considering that food consumption in cities is centrally linked to 79% of all produced food [37], the changing demand for food is linked to the urbanization of food [30]. This will of course affect rural areas and agricultural supply chains [38,39].

The understanding of how food is manufactured and consumed comprises one of the main aspects of urbanized justifiable expansion and food security but also affects rural areas, in relation to CC and socioeconomic inequalities [40]. Moreover, the globalization of the 1980s led to the increased disconnection of cities from food [41]. Hence, food systems are managed at the national level, since urban policies and regulations do not often pay significant attention [42,43].

Food security is a major urban problem in developed and high-income countries and around 50 million urban dwellers were found to be food-insecure in 2015, in North America and Europe [44]. This now includes food accessibility.

Cities rely on external markets and long food chains, hence being vulnerable to supply chain shocks, including CC [45] or pandemics [46].

Recently, [47] stated that the key actors towards more sustainable food systems, despite the lack of a clear mandate, are city governments (and territorial communities) [48].

Food system transformation can be defined as ‘a process of major and key change in the food system structural, functional and relational issues leading to more equitable relationships and more benign patterns of interactions and outcomes’ [49,50].

Enhanced participatory governance structures using a multi-actor approach can be achieved with cities playing a pivotal role, according to Mattioni et al. [51]. National governments, due to their capacity to invest resources in the food system infrastructure, should promote food system transformation beyond local areas to create cohesion [52].

Eight projects contribute to the Food 2030 priority of nutrition and sustainable and healthy diets, nine projects contribute to the Food 2030 priority of the climate and environment, and twelve projects contribute to the Food 2030 priority of innovation through empowering communities.

A useful framework based on place-based solutions, the connection of food with the climate and community, and the circularity and diversity of approaches is the recently developed Client-Led Information System Creation (CLIC) framework [53]. CLIC stands for ‘conceptual framework for integrated food policies and intervention design’ and is conceptualized by four pillars:co-benefits across social, environmental, and economic objectives;linkages between rural and urban areas;the inclusion of all stakeholders and their knowledge;connectivity between food and other policy priorities (e.g., Food 2030).

## 4. Food from Ocean and Freshwater Resources

A key factor for European and global food and nutrition security is seafood production through harvesting (fisheries) and farming (aquaculture). Primary food production systems contributing to food and nutrition security by 2030 comprise sustainable fisheries and aquaculture [54].

Europeans consume roughly twice as much as they produce [55] and most imports come from Asian countries.

By 2030, aquaculture could enhance seafood production and deliver close to two thirds of the global seafood demand [56]. However, this necessitates development in sustainable and less impactful ways, including freshwater aquaculture, which is decisive for noncoastal countries, as reported in the Blue Growth Strategy.

Sustainably farmed seafood production requires overcoming obstacles such as a lack of knowledge of the elementary biology/ecology of fish and shellfish, sickness prevention, and management. Moreover, there is a need to control weak governance structures for fisheries management and build on new technology uptake by the fisheries sector, as well as the consumer acceptance of farmed seafood. Hence, better risk assessment and management in seafood systems will be required [1].

Approximately three billion people are supplied with fish, with an average per capita animal protein intake of 20%, accompanying various crucial micronutrients. Around 10–12% of the world’s population depends on blue foods for their livelihoods [57].

Operational costs for aquaculture producers, seafood processors, and fishermen come from energy and raw material price increases, according to Rahman et al. [58]. Prices are reaching EUR 1 per liter in several EU nations, with the industry claiming profits from EU vessel operations of up to EUR 0.60 per liter [59,60].

The worldwide transition to a sustainable agri-food system will be supported by the EU. This will include the sustainable management of fish and seafood resources and the control of ocean governance, marine cooperation, and coastal management. Illegal, unreported, and unregulated fisheries will face a zero tolerance policy. The governance of the agriculture and fishery industries can enhance the cycle of sustainable development for food and nutrition security, as reported by [61,62,63,64]. This will affect global food security and development.

## 5. Alternative Proteins and Dietary Shift with a Focus on Biological and Chemical Modification of Plant-Based Proteins as a New Sustainable Solution

Changes in ecosystem services have resulted from heavy industrialization and agricultural intensification (excessive use of fertilizers and pesticides), which have radically changed the N and P cycles, indispensable for plant growth. Hence, if water is polluted, this will affect soil productivity [65].

The application of a reasonable reduction in animal-based calorific consumption can reduce emissions from agricultural production, as stated by the Commission’s Communication, ‘A Clean Planet for All’ [66].

The Intergovernmental Panel on Climate Change (IPCC) [67] stated that composed diets (plant-/animal-based foods) can achieve climate change adaptation and mitigation and this can be beneficial for human health [68]. Moreover, one fifth of the change requested to limit warming to <2 °C could arise from dietary shifts.

Milford et al. [69] suggested the indirect interrelationship of consumers’ preferences and consumption habits. Castellani et al. [70] showed that dietary shifts towards less animal-based food, resulting in a decrease in the environmental impact of food consumption, will be affected by marketing issues. Consumers’ choices and food production will affect the linkage between the food supply and diets [1]. Hence, research on the improvement of the processing of alternative proteins is imperative, taking into account biotechnology pathways that provide nutritional and sensorial food quality coupled with environmental sustainability.

Regarding plant-based proteins, a focus on biological and chemical modification is described, since it is a means to improve them and contribute to their sustainability and full utilization.

Proteins offer a range of amino acids that are essential in preserving human health, and, in food technology, they provide some functional properties, e.g., as stabilizing, emulsifying, thickening, gelling, foaming, and binding agents [71,72,73]. Biochemically, proteins’ capability to retain their functional traits relies on their intrinsic structure, their shape/configuration, and how they interrelate with food constituents [74]. Nowadays, the processing and exploitation of plant proteins have attracted universal attention and numerous scientific investigations are focused on the improvement of the application of plant proteins in the pharmaceutical and food industries through modification techniques. The latter could increase their techno-functional aspects, bioavailability, bioactivity, and digestibility characteristics [75,76,77].

Biological and conventional food processing changes to food proteins are easy to implement and have been extensively employed. Current chemical protein strategies for site-selective modification can be mastered by adding new, efficient moieties or the exclusion of components from the protein structure. Numerous chemical and biological modification approaches will be discussed in this section, with examples contributed precisely in the plant-based protein context. Martinez-Alvarenga et al. [78] produced glycoproteins with maltodextrins (MD) attached per whey protein isolate (WPI). Their solubility was augmented at the pI by combining them with MD. These authors demonstrated that the increase in the foaming capability and foam stability of WPI after glycation with maltodextrin was linked. In the same way, emulsification was improved and presented monomodal performance due to the developed steric revulsion [79]. In this study, the authors demonstrated the potential of the electrospinning technique in the modification of pea protein isolate and as a pretreatment to facilitate the production of Maillard conjugates with improved functionality. In another study, oat protein was glycated with dextran and β-glucan, which enhanced its solubility and emulsification potential via alterations in its secondary structure, leading to the random coiling of the protein conjugate [80]. The conjugated product displayed elevated viscosity, attributed to the functional properties of oat β-glucan and its concentration. In the Meng et al. study, it was reported that the Maillard reaction enhanced the solubility and emulsion capacity of rice dreg protein (RDP) [81]. It was reported that the modified protein conjugates had improved immunomodulatory properties when examined using a cyclophosphamide-induced immunodeficiency animal model. To improve the thermal stability of canola protein isolate, Maillard glycation was effectively applied by Pirestani et al. [82]. Through Maillard reactions in aqueous solutions, Xue et al. [83] revealed that buckwheat protein isolates reacted with dextran to produce conjugates and this reaction was enhanced with ultrasound. Structurally, more random coils and less α-helices in the conjugates were generated. The conjugates presented comparable solubility behavior and improved emulsification and thermal stability. The functional properties of food proteins can thus be increased by several modifications.

Amongst all chemical modifications, phosphorylation has been established as an effective method of enhancing the functional properties of proteins like potato protein isolate (PPI). At pH 8, phosphorylated products (PP-PPI) with sodium trimetaphosphate (STMP) were marked by the maximum oil absorption and foam capacity and emulsion activity. At pH 10.5, WHC was superior to the native ones [84]. Similarly, Sánchez-Reséndiz et al. [85] demonstrated the improved emulsifying activity of peanut and soy protein isolates after phosphorylation using STMP. In this study, peanut protein isolates’ phosphorylation changed greatly the values of the emulsifying activity (+6.6) and in vitro protein digestibility (1%). Remarkably, in soybean, all functional properties were improved, excluding the water solubility index (WSI) and foam activity (FA). The thermal aggregation and viscoelasticity of rice glutelin (RG) were also enhanced after STMP phosphate modification, as stated in the study of Wang et al. [86]. These authors concluded that the generation of protein aggregates was credited to the interactions between proteins, comprising covalent and non-covalent interactions. The particle size distribution, intrinsic fluorescence emission spectra, surface hydrophobicity, and rheological behavior results indicated that three steps were realized: (i) the RG was unfolded; (ii) the unfolded proteins created oxidized free SH groups to form disulfide bonds and hydrophobic interactions; and (iii) a 3D system structure was formed. The modification of the secondary structures through heat treatment presented an increase in β-sheets and a decrease in α-helix content, leading to the generation of protein aggregates [86].

Protein acylation between the amino acid residues of proteins and anhydrides is one type of chemical modification that serves to increase the function and physicochemical properties of food proteins. Zhao et al. [87] found that the addition of succinic anhydride to oat proteins apparently improved their gel properties compared to the unmodified oat protein. After acylation, the gelling property of rapeseed protein achieved additional functionality, as assessed by Chen et al. [88]. Heightened thermal stability, solubility, and emulsifying properties were also reported due to succinylation-induced changes in the secondary structure of male date palm pollen protein concentrate [89].

In contrast to acylation and alkylation, involving chemicals, deamidation can be achieved under moderate conditions and without additional molecules. Since legumes/cereals have very high proportions of Gln and Asn, deamidation can be a suitable tool to apply them in the food industry using several approaches (such as alkali, acid, and enzyme). For instance, by using acids such as acetic acid, citric acid, and tartaric acid, the bitterness of wheat gluten hydrolysates was masked [90]. To improve the solubility of rice bran protein, Guan et al. [91] used alkaline deamidation (pH 12 and 120 °C for 15–30 min). For enzymatic deamidation, it was reported that GTase was the most investigated enzyme in the deamidation of plant-based proteins. Through GTase deamidation, Hadidi et al. [92] improved the WHC, solubility, and emulsifying and foaming properties of *Oenothera biennis* L. protein; in the same way, a techno-functionality improvement in pea protein isolates was observed when GTase deamidation was utilized [93]. Industrially, it should be noted that only deamidase isolated from *Chryseobacterium proteolyticum* has been applied [94].

Biological modification, linking enzymatic and fermentation approaches, is an alternative modification technique that is eco-friendly, less energy-consuming, and free from the production of toxic by-products. Regarding the enzymatic approach, the class of the employed enzyme plays a central role in the final features of the modified plant proteins, considering that their molecular changes translate into different specific cleavage sites [94]. For example, tryptic hydrolysis, especially at pH 4, resulted in the formation of oat protein peptides with an improved, homogenous foam structure, a rapid foaming ability, and a highly viscoelastic interfacial film [95]. In addition, pepsin can generate hydrophobic peptides with high hydrophobicity and surface-active properties [96]. These authors observed the improved WHC and OHC of pea protein-enriched flour after pepsin hydrolysis. Sun et al. [97] prepared enzymatically modified walnut dregs (CPMP), based on protease, and indicated that the generated product showed interesting emulsifying and foaming properties. On the other hand, fermentation has been used as a biological tool for plant-based protein modification. In this sense, various starter cultures have been used for the fermentation of plant proteins, such as lactic acid bacteria (LAB), mold, yeast, and *Bacillus* strains. Fermentation, using Lactobacilli strains, was recorded to enhance soy protein’s solubility, WHC and OHC, and foaming properties [98,99]. Similar trends were reported by Klupsaite et al. [100], who used *Pediococcus pentosaceus* KTU05-9 in the solid state fermentation of lupin protein. Interestingly, fermentation can significantly decrease the beany and bitter off-flavors of different plant-based proteins [101]. These authors used eight LAB starins, and *Lactobacillus brevis* had strong potential to improve the techno-functional properties of lupin protein. A summary of the above chemical and biological modifications is given in Table 1.

## 6. Resource-Efficient Food Systems and Food Waste

Accelerated action to reduce food loss and waste represents United Nations Sustainable Development Goal 12.3, with the target of halving FW by 2030 [110]. “A reduction in food quantity and quality” is the definition of food loss and waste [111] and refers to “food lost or wasted in the part of food chains leading to edible products going to human consumption” [112]. Fish provides 20% of the average per capita intake of animal protein, making fisheries central to achieving food security [113] in order to feed more than 3.3 billion people globally [114]. All food losses take place along the food supply chain (FSC) and the retail level is included through FW [1]. Two separate indexes, the Food Loss Index (FLI) and the Food Waste Index (FWI), have been reported by the FAO and the United Nations Environment Program (UNEP). It is projected by the FLI that 14% of food produced is lost from post-harvest without retail [115,116].

Drivers of FW are differentiated as below.

Specific food products are represented by generic and systemic approaches [117]. The highest FW generation (46%) comes from the consumption stage, succeeded by principal production and processing/production at 25 and 24%, respectively.

Regarding FLW, according to the World Food Programme [118], hunger is a common paradox as it leads to food insecurity. In 2019 alone, the EPA estimated [119] that the food retail, food service, and residential sectors accounted for approximately 66 million tons of wasted food, with most of this waste (about 60%) directed to landfills. The EPA estimated that, in 2018 in the U.S., 24 percent of the amount was landfilled and 22 percent of the amount was combusted with energy recovery. Moreover, they reported that landfills and combustion facilities were overloaded with more food than any other single material [120].

Reducing wasted food saves resources such as land, water, energy, and labor; reduces greenhouse gas emissions (the majority of greenhouse gas emissions from wasted food result from production, transport, processing, and distribution); and reduces methane from landfills. It is necessary to consider that when wasted food enters landfill, the nutrients in the food never return to the soil. The EPA estimates that 58% of landfill methane emissions to the atmosphere come from wasted food [121]. Nutrients can be returned to the soil by composting, hence supporting a circular economy (CE).

In 2021, the EPA released the first of two reports in a series on the environmental impacts of wasted food. Part 1 was titled From Farm to Kitchen: The Environmental Impacts of U.S. Food Waste (https://www.epa.gov/land-research/farm-kitchen-environmental-impacts-us-food-waste [120]) (accessed on 28 December 2023), and Part 2 was titled From Field to Bin: The Environmental Impacts of U.S. Food Waste Management Pathways [121], released in 2023.

The most important themes in discussing FW nowadays are as follows: (1) the anaerobic digestion of FW for CE conception; (2) FW systems and life cycle valuations for CE; (3) bio-based CE methods; (4) consumer performance and approaches towards CE; (5) food supply chains/FW in a CE; (6) material flow analysis and sustainability; (7) challenges, policies, and practices involved in achieving circularity; and (8) CE and outlines of consumption [122].

The CE is designed to substitute traditional linear supply chains with systems in which materials are recycled within creation systems, grounded on the principle of “waste = food” [123,124], reinforcing the transition from recycling to upcycling [125]. This refers to any waste transformation process transforming waste into higher-value products by using them as input for other products. Hence, the CE seeks to transform one person’s waste into another person’s resources [126], stimulating radical innovation and integrating human activities into ecosystems [127]. International policy makers often discuss the need to shift towards a CE [128]. However, in order to allow the CE to shift towards sustainability, several actors need to be engaged [118], including society and consumers [129].

A food use hierarchy should be employed focusing on prevention, followed by the redistribution and reprocessing of surplus food to people in need, the production of animal feed, and recycling and disposal, as shown in Figure 3 [130].

A Malaysian model has been proposed analyzing the factors affecting FW [131]. Poor food management practices and gender are important issues that affect FW, reinforced by the consumer behavior concept. The projected presented rice waste in the CE model, which was well accepted by the public. Moreover, they mentioned people’s readiness to pay a certain sum of money to process their FW.

A study conducted in Daegu, South Korea in 2019–2020, collecting FW from 218 households, showed an average daily contribution of FW of 0.73 kg per household, with the equivalent greenhouse gas emissions of 0.71 CO_2_, a water footprint of 0.46 m^3^, and economic losses of KRW 3855.93 [132].

## 7. The Microbiome

Relevant EU policies in this direction involve energy security and climate change, climate mitigation [133], and adaption to CC, referring to the new bioeconomy strategy [134].

Beneficial symbiosis provided by certain microbiomes can lead to higher yields and nutritious food. The Soil Health and Food mission [135] requires sustainable and circular management, the use of natural resources, and soil health improvements.

A possible agronomic management measure for crops is the exploitation and exploration of the microbiota in agroecosystems in symbiosis with crop plants. Multiple benefits are underlined by this innovative strategy, such as coping with biotic and abiotic stresses—for instance, heat stress [136,137], drought [138,139], waterlogging [140,141], and the attack of pests [142] and diseases [143,144].

## 8. Nutrition and Sustainable Healthy Diets

Major challenges in global and EU food systems involve all aspects of nutrition, including malnutrition (undernutrition, over-nutrition, and micronutrient deficiencies), as well as environmental issues such as climate change and resource scarcity, in connection with urbanization and food poverty.

According to IPES FOOD [145], approximately 2 billion people are overweight or obese, 2 billion are affected by micronutrient deficiencies, and 800 million people are undernourished.

A leading non-communicable disease (NCD) hazard feature and a principal cause of obesity is an unmaintainable and unnatural diet, with an augmented demand for livestock products, along with the consumption of (ultra)processed food that are high in calories, with no nutrients [146].

The reduction of obesity rates over the last few decades has not been accomplished by any EU country to a significant extent [147]. In Europe, at present, children and the poorest individuals are severely affected, with more than half of the adult population being overweight or obese [1].

The adoption of long-term healthy and sustainable diets and the reduction of non-communicable diseases (NCDs) are representative innovative solutions to achieve optimal health and well-being. A more thorough understanding of the relation between lifestyle (including nutrition and alcohol), geographical (national/regional and rural/urban zone), and environmental factors is required, as well as assessing biological parameters and the risk of NCDs.

Consumer choice is affected by consumer behavior, and dietary behavior is not only impacted from birth, but also by food environments, policies, gender, and nutrition labeling [1]. Food choices, lifestyle, motivation, and decision-making are heavily affected by human neurobiological pathways. Hence, people should modify their dietary models and make healthier selections. In this direction, the combination of data from different domains and artificial intelligence needs to be employed towards the study of consumer behavior and dietary intake, as well as their impact on health and environmental sustainability. In addition, the optimization of nutritional, structural, and functional food properties from new raw materials or organic ones with a low impact on the environment and in accordance with new style food consumption preferences should be implemented [1].

A One Health approach should be adopted, showing the links among the health/well-being of people–animals–plants–environment, that sustains their existence [148], optimizing the health of people, animals, and ecosystems [149]. This includes nutritious food [150].

Threats to “life on Earth” have been addressed by the Manhattan Principles [151]. They introduced 12 maintainable methods to evade and avoid epidemic or epizootic diseases [151,152]. Some actions require holistic expertise [153,154,155,156,157]. In this context, potential regional differences regarding One Health research priorities were observed by [158]. Potential regional gaps and differences in One Health research priorities were highlighted, emphasizing the surveillance versus policy activities in One Health.

Global food systems should be able to provide access to diets that allow adolescents to grow and thrive and not to fail [159]. Access to nutritious, safe, affordable, and sustainable foods for all adolescents should be critical [160]. Dietary intake should be improved for most adolescents around the world and not be based on energy-dense, nutrient-poor processed and ultra-processed foods [159,161] with low intake of fruits, vegetables, and whole grains. Another reason is that ultra-processed foods have a higher environmental impact and affect the sustainability of dietary patterns [162].

Despite this, sustainable adolescent nutrition should cover the large gaps in dietary intervention, policy, and programming that still remain [163]. Their complex food environments should be explored further, investigating their decisions regarding what to eat, where to eat, and where to buy food [164]. Adolescent policy and practice initiatives should be adopted and implemented effectively [165,166].

Moreover, the basis of all policy design should be explicitly adolescent-centered and designed to co-construct knowledge [167,168,169,170].

Nutritional life cycle assessment (n-LCA) can be used to measure environmental impacts against the nutritional value of food levels (i.e., production systems, food items, and diets/food supply). It can use nutrient or health metrics in the impact assessment phase, as carried out in the CONE-LCA framework, reported by Stylianou et al. [171].

An illustration of milk and dairy production in the food economy system and the food chain is represented by the attitudes of a group of young Polish consumers towards selected features of dairy products [172]. Consumer opinion surveys include key elements of improving the food market and assessing consumers’ approaches to current issues related to access to high-quality food. Gaworski et al.’s [172] investigation aimed to determine the attitudes of young Polish consumers about dairy products. The objective was linked to the evaluation of selected features of dairy products and their packaging and the assessment of regional products and novelties in dairy production. These consumers pointed out the importance of the quality/taste of dairy products and the minor role of packaging. Additionally, most respondents claimed that they did not notice to the biodegradability of the packaging. When asked about regional dairy products, respondents paid great attention to their value, resulting from natural methods of production, without preservatives. However, a small number of young respondents showed knowledge of the idea of dairy production ‘from grass to glass’, which would indicate inadequate interest in innovative solutions in the dairy sector.

Nowadays, the most comprehensive assessment of nutrient metrics in LCA is the recent FAO paper by McLaren [173], which establishes a basic foundation and high-level overview. Other papers on LCA include [174,175,176,177].

The nutritional and environmental contributions of selected production practices can be measured by farmers, who receive higher prices for foods with a stronger sustainability profile [178,179].

Green et al. [180] also highlighted the inclusion of capping, weighting, energy standardization, across-the-board versus group-specific aspects, dietary- and/or context-specific aspects, validation, disqualifying nutrients (e.g., saturated fat), reference amounts, processing quality, the selection of nutrients/ingredients, interpretation, and data quality. The largest impact was attributed to energy standardization and dietary specificities when assessing nutrient indices in isolation, as well as capping and disqualifying nutrients. Spearman rank correlations and Wilcoxon signed rank *p*-values were also calculated.

## 9. Food Safety Systems of the Future

To become a climate-neutral continent by 2050, the EU has promoted the Green Deal, with the Farm to Fork and the Biodiversity strategies as its foundations [181]. The goal of the Farm to Fork strategy aims to transition to a maintainable food system [182]. The Biodiversity strategy involves numerous proposed policy actions with the goal of lower greenhouse gas emissions, biodiversity conservation, reduced pesticide use, and augmented consumer empowerment [183]. The Farm to Fork strategy predicts the creation of a more justifiable food system by applying restrictions on fertilizer and pesticide use and placing at least 25% of the EU’s agricultural land under organic farming by 2030 [182]. The Farm to Fork strategy and the Biodiversity strategy aim to enhance European biodiversity levels by increasing the agricultural land amount under high-diversity landscape features to at least 10% [182]. However, there are concerns regarding whether organic production and certification are adequate to preserve and expand biodiversity and to achieve the climate targets as defined in the UN Sustainable Development Goals. Therefore, holistic approaches that go beyond organic production, which also address the role of consumers in food labeling, are suggested [184]. In this context, the ecological food system framework initiative was introduced, with the goal of making the EU food system sustainable and integrating sustainability into all food-related policies [182]. While progress has been made in quantifying product-specific environmental impacts, major challenges in how to track and communicate these influences continue. For instance, there is a lack of EU-wide monitoring frameworks to appraise progress towards food sustainability objectives. An initiative for green claims was launched to substantiate green product credentials against a regular methodology to evaluate their impact on the environment.

An integral part of food and nutrition security (FNS) that is very important and significant for health and a sustainable environment is food safety. The Rapid Alert System of Food and Feed (RASFF) and legislation from farm to fork focus on ensuring a high level of food safety and animal health and welfare and plant health in Europe. The European Food Safety Authority (EFSA), responsible for scientific advice, is developing new risk assessment methodologies for emerging new foods and existing ones. Despite the strong focus on food safety, there is limited evidence of an integrated systemic approach to FNS, leading to the decentralization of food policy [185,186]. Food safety is part of a sustainable food system, minimizing the risk of the transmission of toxins or pathogens through the food system and minimizing the use of antibiotics, pesticides, and other substances of concern. Complex interactions due to CC with a number of food safety hazards lead to increased risks of foodborne illnesses and affect safe and nutritious food for millions of people around the globe, and these should be considered and addressed by innovative global food security systems. Food security and livelihoods could be at risk despite the lack of connectivity of the current COVID-19 pandemic to food safety in the EU. Proactivity regarding biological hazards and, in the case of the food system, those that emerge throughout the food chain remains imperative. Food safety risks might also be increased by changes in food and farming systems. According to the recently published FAO document on CC and food safety [187,188,189,190,191], the understanding of CC and novel approaches and applications should be furthered. Emerging technologies in various areas of the food chain should be further strengthened among all actors and stakeholders. This will lead to innovation and provide solutions to address food chain challenges in association with consumers.

## 10. Food Systems in Africa

Meta-analyses suggest that mobile devices disseminating agricultural information in sub-Saharan Africa and India have improved yields by 4% and the adoption of agrochemical inputs has increased by 22% [192].

Better nutrition performance of African farming systems should be sought, thus strengthening the link between agro-biodiversity, aquaculture systems, and food diversity. Technological, food safety, social, and gender issues should be taken into account regarding local food systems. These will include sustainable post-harvest technologies and bio-based packaging approaches, for the reduction of food waste. The diversity of diets and improvement of food identity will be derived from small farmers and processors, benefitting rural areas. This means that a focus on food supplies for local urban markets and high-value global markets is essential. Diet-related, non-communicable diseases and persistent undernutrition are some of the most commonly reported nutritional imbalances in both Europe and Africa. The UN predicts that the global population will increase from 7 billion to >9 billion by 2050 [193], of which the majority will reside in Africa. Nutrition performance needs to be linked with sustainable agricultural systems, thus strengthening the agro-biodiversity of resilient cropping systems. This will lead to a more balanced, healthy diet in order to satisfy population growth and address challenges associated with enhanced climate change. Furthermore, the development of resource-efficient, resilient food value chains should deliver sufficient, safe, affordable, and nutritious food to local consumers and for high-value global markets. Africa’s wealth of local varieties, food intelligence, and healthy diets, including plant-based proteins, should be exploited and explored, as they are currently not considered and not reaching the market.

In the framework of the SDGs, the EU–Africa RI Partnership on FNSSA proposals describe how projects can contribute to

sustainable, healthy African diets based on sustainable and secure food systems (comparable to the Mediterranean diet);the sustainable growth of food chain operators (SMEs) in rural areas in Africa, along with the involvement of small farmers (including aquafarmers);novel food products, tools, and processes applicable in Africa, addressing food safety issues across the entire food value chain;the implementation of nutritional recommendations leading to significant reductions in childhood malnutrition in Africa;the founding of the EU–Africa Research and Innovation Partnership on FNSSA and influence at a local level;pilot modernization activities for the benefit of African and European consumers.

InnoFoodAfrica [194] focuses on nutrition, with the overall objective of improving the nutrition and well-being of African people via the analysis of diets, growing a wider variety of crops, and developing new food products and ingredients and new food processing technologies. This is complemented by the use of crop side streams to produce biodegradable composite materials for packaging and other end uses.

To obtain a deeper understanding of the nutritional situation in several African cities, a survey was conducted on sub-Saharan diets. Through the survey results, nutrient gaps were analyzed and the authors created recommendations to close these gaps. Here, the Food 2030 priorities become very tangible: affordability needs to be a key element in relation to nutrient availability, ensuring that healthy diets are affordable. Some investigations have been conducted that indicate how the aquaculture and fishery industries are managed in Africa as a means of guaranteeing food security and economic growth [195,196,197,198].

## 11. Data and Digital Transformation

The economy and society are being transformed by digital technologies. Data are growing and expected to increase significantly by the year 2030. Newly evolved technologies such as artificial intelligence and new smart-connected objects (e.g., farm machinery, robotics, home appliances, wearables) have led to this data increase, along with data processing and analysis [1,199].

Enormous benefits for citizens, farmers, food businesses, researchers, and society have been brought by this data-driven innovation.

Digital technologies have entered the whole food supply chain and allow us to address techno-economic challenges in the agro-food sector. Digitalization in agricultural and food systems, introducing innovative tools and systems, can help in the achievement of the 2030 United Nations Sustainable Development Goals.

In Sridhar et al. [200], the application of digitalization in the agriculture and food sectors has been reviewed. Artificial intelligence, precision farming, and big data analytics have been employed and applied in agriculture. Environmental, social, and economic sustainability would be the end result of the integration of these techniques. A 23% reduction in costs and a nearly 5% decrease in the volume of medium-level waste have resulted from the application of these digital technologies in agriculture.

The amalgamation of these smart systems and technologies in the agricultural and food sectors might affect their productivity and output, since they allow the highly effective storage, transmission, and retrieval of electronic data.

All sustainable goals will be affected by the application of digitalization in the agro-food sector; however, the main pillars are related to SDG 2, which centers on eradicating hunger and achieving food security, and SDG 1 (no poverty), focused on improving access to markets, financial services, and valuable agronomic information. The final outcome will be the enhancement of the livelihoods of smallholder farmers and marginalized communities. Real-time monitoring and traceability, thereby reducing foodborne illnesses and making food safer, in the context of SDG 3 (good health and well-being), will also be beneficial due to the application of digital technologies.

SDG 6 (clean water and sanitation and improved water resource management and conservation) will be satisfied by the use of digital technologies in agriculture. SDG 8 (decent work and economic growth), which requires the creation of employment opportunities within the agro-food tech sector and the promotion of sustainable economic growth will also be affected by the application of digitalization.

SDG 9 (industry, innovation, and infrastructure) will also benefit from innovative agro-food technologies, through improving the production and distribution infrastructure.

SDG 12 (responsible consumption and production), focusing on resource-efficient digitalized farming practices, will be affected, hence reducing food waste and enhancing traceability in supply chains.

Digitalization will also affect climate resilience and mitigation effects, hence supporting SDG 13 (climate action). SDG 14 (life below water) will also be affected due to more sustainable marine ecosystems [200].

The optimization of seafood farming practices can be achieved by digitalization, thereby reducing the strain on marine resources and aiding the conservation of aquatic life.

Furthermore, sustainable land use and agriculture, as achieved with the use of digital technologies, aligns with SDG 15 (life on land). The minimization of the use of harmful agrochemicals, the reduction of soil erosion, and the encouragement of sustainable land management, affecting the conservation of terrestrial ecosystems and biodiversity, can be achieved by data analytics.

Industry 4.0 is a new era with biofortification. The enhancement of the nutritional content of crops will be carried out, directly aligning with SDG 2.2 and attaining SDG 2.3 through the improvement of agricultural productivity. Ensuring sustainable food production systems will also contribute to SDG 2.4. Sensors could also be used along with hydroponics systems and vertical farming [201].

Artificial intelligence (AI) connects three subfields: machine learning, deep learning, and neural networks. Algorithms for statistical predictions and inferences, such as image recognition, are employed in machine learning [202]. The connection of information arises from deep learning via the use of various datasets or nodes. Speech recognition is accomplished by deep learning. Neural networks, which leverage statistical models and algorithms, can predict data and recognize patterns [203]. The prediction of weather conditions and soil patterns essential for the growth of crops can be achieved by employing these technologies.

Big data involve the five Vs, namely volume, variety, velocity, veracity, and value [204]. Volatility, validity, visualization, variability, vulnerability, visibility, and vagueness are also considered [205]. Robots, satellite imaging, remote sensing, and geospatial data are being used the most in understanding crops and performing land mapping and soil management [206,207,208,209,210,211,212].

Several barriers exist that limit the potential of data-driven innovation.

**Lack of data governance:** missing organizational approaches and structures.

**Imbalances in market power:** a large amount of data is accumulated by a small number of players (e.g., food retailers, data aggregators like Google or Facebook, equipment manufacturers, ERP platforms). The implementation of new approaches (towards SMEs, farmers, and citizens) will tackle the oligopolistic characteristics of today’s data economy [213,214].

**Empowerment of individuals to exercise their ‘data rights’:** the generation of large amounts of data might lead to risks of discrimination, unfair practices, and lock-in effects. GDPR and ePrivacy access at a granular level should be granted [215].

**Low availability of data:** the scarcity of data available for innovative reuse and low awareness of benefits of data sharing. Some of these data could be part of a ‘data commons’ for EU food systems.**Low awareness of the potential of data-driven innovation and low skills and low uptake:** in EU food systems, cost-effective solutions at the farm and post-farm level should be tested and demonstrated [216]. The path to digitalization is not clear and transparent to many of them [217], which negatively affects the uptake of data-driven innovations. There is a delay in digitalization among SMEs in general and the difference between SMEs and large firms is shown in the 2020 EU SME strategy. Only 17% of SMEs have employed digital technologies in their businesses, whereas large companies represent a higher percentage of 54% [218].**Lack of data infrastructure and technologies and lack of cybersecurity:** cloud services, allowing secure, energy-efficient, affordable, and high-quality data processing processes, will aid the digital transformation process.

## 12. Zero Pollution Food Systems

CC is threatening global crop production. A major cause of global yield losses is the rising temperature [219,220,221], especially in less developed and warm areas such as sub-Saharan Africa and Latin America [222]. Exposure and vulnerability to CC can be reduced by strengthening the adaptations of agricultural systems [223]. Cultivar shifts and agronomic management practices to enhance adaptation have been employed [224,225,226,227]. Cultivars (with higher heat tolerance) and management (e.g., irrigation) adaptations could reduce yield losses due to warming by ~5% in the mid-21st century [227]. Innovative adaptation pathways might include soil-based strategies [223,227] to address climate risks in agricultural systems.

Improving soil health and resilience is fundamental for sustainable food production; in particular, the role of soil in maintaining or improving global crop productivity under climate warming has been identified and quantified by Deng et al. [228].

Since the 1990s, the push towards decarbonization in the chemical industries (e.g., cosmetic, pharmaceutics, and food) coupled with that of the energy sector is imperative. As proposed by de Boer and Van Ittersum [229], the bioeconomy aims to reduce food and organic waste, using biomass as a source of energy as well as feed and then upcycling organic by-products and waste (manures, agri-food wastes, etc.). The reduction of nitrogen (N) inputs and the development of agroecological practices is a means to stabilize or decrease agriculture GHG emissions, as carried out by several countries [230,231].

## 13. Conclusions

The importance of science–policy–society interfaces (SPSIs) is outlined by the transformation of food systems by connecting science, policy, and society. On the other hand, research and innovation might be a strategic driver in the transformation to more sustainable food systems and thus characterizes a key planned area in the Farm to Fork strategy and EU Green Deal. In this way, all stakeholders should not only be involved in discussing the interpretation of evidence but also in the process of determining the action pathways to pursue towards quality, security, and safety. In addition, capitalization and underwriting for the development of skills and tools for research and innovation among food system stakeholders to achieve food system transformation is vital in supporting this transformation.

Referring to this gap, the current review describes tools to improve the capabilities among food system stakeholders in terms of understanding both how food system transformation has ensued and how the required transformation is restricted. Moreover, an extensive discussion is presented on the implications for food system transformation in terms of nutrition and sustainable healthy diets. The latter are needed to achieve changes in the food safety systems of the future. The linkage of food and the environment is evident, and the focus on nutrition and a sustainable healthy diet is well established.

Governance and system change, food security and safety, system transformation, and sustainability transitions have been well analyzed and underlined. All have their significance and the end result is manifested in how consumers will interact with these systems and how their lives will be changed by the consumption of a more nutritious diet. Of course, it is crucial that the environment is not further affected negatively. In this direction, a dietary shift towards alternative proteins and, more specifically, plant-based proteins is imperative. Hence, the technical aspect of their biological and chemical modification has been critically analyzed. This paper will be of value to all readers, whether they work with food, health, or in the food economy system, enabling them to apply and integrate the tools for transformation and adopt more inclusive, transdisciplinary, and systemic approaches to the demanding challenges that we face today.

## Figures and Tables

**Figure 1 foods-13-00306-f001:**
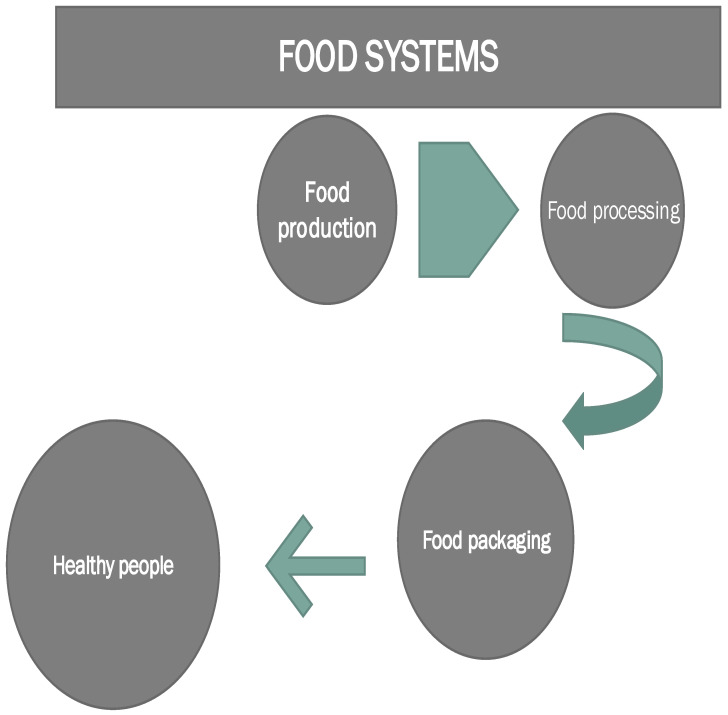
Research and innovation for future-proofing of food systems (adapted from European Commission, 2020).

**Figure 2 foods-13-00306-f002:**
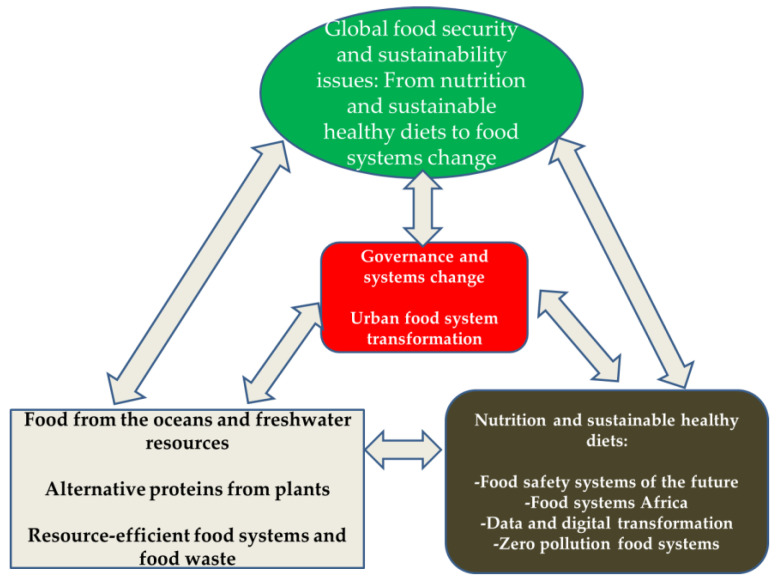
Flowchart underlying governance, food system transformation, and sustainability transitions.

**Figure 3 foods-13-00306-f003:**
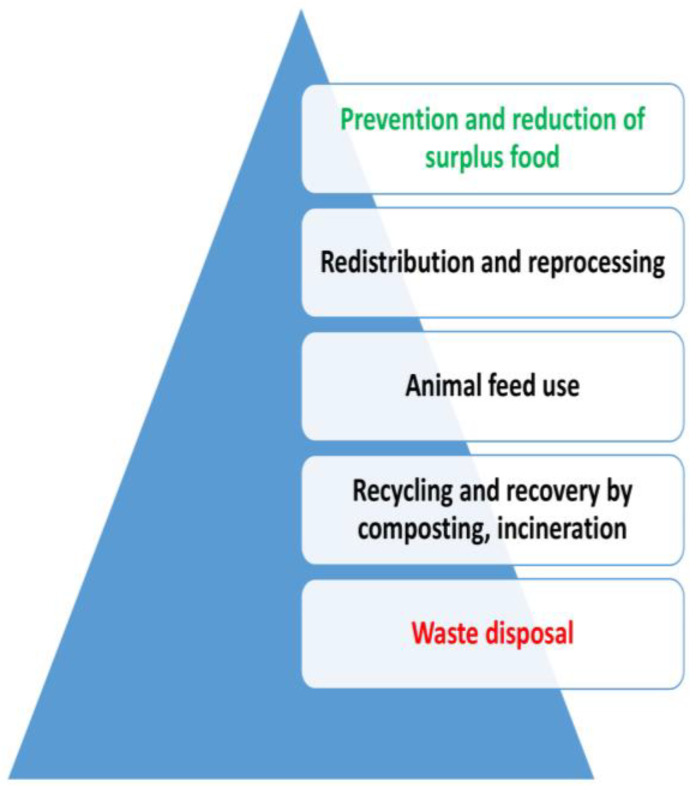
Sequence of management of food surplus, by-products, and FW deterrence: 10 strategies. Adapted from [130].

**Table 1 foods-13-00306-t001:** A summary of the chemical and biological modification approaches for plant-based proteins.

Modification Strategy	Methods/Reactions	Source of Plant Protein	Techno-Functional Properties	References
Chemical	Deamidation—alkaline	Evening primrose (*Oenothera biennis* L.)	- Improvement in functional traits - Edible protein was generated	[92]
	Deamidation—glutaminase	Pea	- Improvement in solubility - Improvement in techno-functionality characteristics of pea protein isolate - Increase of beany flavor, grittiness and lumpiness.	[93]
	Acylation and additional transglutaminase catalysis	Rapeseed	- Improvement in gelation properties - Rapeseed protein isolate (RPI) had good thermal stability, gel strength, apparent viscosity, and surface roughness	[88]
	Glycation by electrospun fiber-assisted drying	Pea	- The emulsion stability and solubility of pea protein hydrolysate were improved	[79]
	Glycosylation (microwave-assisted wetting)	Rice	- The solubility was increased - With Maillard reaction, the emulsifying capacity was enhanced - Good immunomodulatory properties of rice dreg protein	[81]
	STMP phosphorylation	Rice glutelin	- The turbidity of phosphorylated rice glutelin (PPRG) was enhanced - The viscoelasticity, phosphorylation, and thermal aggregation of rice gluten were improved	[86]
	Deamidation—alcalase hydrolysis	Wheat	- The bitterness was masked	[102]
	Acylation and glycation	Rapeseed	- Improvement in gelation properties - Improvement in water absorption capacity and textural properties	[103]
	STMP phosphorylation	Soybean and peanut	- Improvement in emulsifying activity - Improvement in in vitro protein digestibility >1%	[85]
	Deamidation	Rice bran protein	- The water solubility was enhanced (=90%) at pH 12 and 120 °C for 15–30 min - Thermal property was preserved	[91]
	Glycosylation (wet)	Canola	- Improvement in viscosity - Improvement in physical structure	[82]
	Glycosylation (ultrasound-assisted drying)	Buckwheat	- Improvement in surfactant capacity - Ultrasonication enhanced the functional properties - Improvement in emulsion stability and solubility	[83]
	Deamidation—proteax/glutaminase SD-C100S	Wheat	- The bitterness was masked	[76]
	Glycation	Oat	- Improvement in emulsification ability and solubility	[72]
	Glycation	Whey	- Improvement in foaming properties and protein functionality	[70]
	Phosphorylation with sodium trimetaphosphate (STMP)	Potato	- Phosphorylation is pH-dependent: (pH 5.2) -Improvement in content of all amino acids of potato protein isolate (PP-PPI) - At pH 10.5, decrease in all content - At pH 8: improvement in oil absorption capacity and emulsion activity and ↗ foam capacity - At pH 10.5: improvement in water absorption capacity	[84]
Biological	Enzymatic: TGase (transglutaminase)	Peanut	- Improvement in the emulsifying activity index (EAI), emulsifying stability index (ESI) - Improvement in gelation and oil-binding properties - Decrease in protein solubility	[104]
	Enzymatic: proteolytic enzymes	Pea	- Improvement in protein solubility at pH 4.5 at all times during hydrolysis - Improvement in foaming with trypsin hydrolysates and emulsifying capacity	[105]
	Enzymatic: papain and pepsin	Pea	- Improvement in WHC and OHC - Decrease in foaming properties and emulsifying properties	[96]
	Enzymatic: pectin methyl esterase	Pea	- Improvement in the degree of esterification - Improvement in solubility	[106]
	Enzymatic by complex proteases	Walnut	- Improvement in solubility - Improvement in water holding capacity - Improvement in emulsifiability and emulsion stability - Decrease in oiliness - No change in foaming features	[97]
	Enzymatic: TGase	Coconut	- Improvement in the mechanical and barrier properties of films based on modified coconut protein	[107]
	Enzymatic: chymotrypsin and protease	Quinoa and amaranth	- Improvement in antioxidant, antimicrobial, and antihemolytic properties	[108]
	Fermentation: *Lactobacilli* strains and *Staphylococcus xylosus*	Lupin	- Improvement in foaming properties and emulsifying properties - At pH 4: no change in solubility - At pH 7, a decrease in solubility and bitterness	[101]
	Fermentation: *Lactobacillus plantarum* strains	Soy	- Improvement in surface hydrophobicity - Emergence of β-strand structure	[99]
	Enzymatic glycosylation: TGase	Black soybean	- Improvement in solubility, rheological properties, and emulsification	[97]
	Fermentation: *Pediococcus pentosaceus* KTU05-9	Lupine	- Improvement in solublity and functional properties, at pH 8.0 - Decrease in bread hardness, chewiness, and resilience - Improvement in bread springiness	[94]
	Fermentation: *Lactobacillus helveticus*	Soy	- Decrease in the beany and bitter off-flavors	[98]
	Fermentation: *Bacillus licheniformis*	Peanut	- Improvement in nutritional properties - Improvement in antioxidant potential	[109]

## Data Availability

No new data were created or analyzed in this study. Data sharing is not applicable to this article.

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
