# Peer review of "Global Food Security and Sustainability Issues: The Road to 2030 from Nutrition and Sustainable Healthy Diets to Food Systems Change"

_foods, 2024, doi:10.3390/foods13020306_

Round 1

Reviewer 1 Report

Comments and Suggestions for Authors

foods-2822232:Global food security and sustainability issues: the road to 2030 from nutrition and sustainable healthy diets to food systems change-a case study of plant-based proteins and their biological and chemical modification

Focusing on global food security issues, this study discussed 11 pathways from farm to fork, trying to find effective guarantees for global food security. The authors seem to have attempted to emphasize the importance of plant-based proteins, but the organization and presentation of the manuscript are not sufficient for this purpose.

Some specific comments are as follows: 

1.     Line 1-5, The title of the article is not enough to convey the full content of the article and is even a bit biased.

2.     Line 7, institution, city and country are not written in a clear order

3.     Line 12-23, the content of the abstract is not centered on the theme or title of the manuscript and does not address the findings, conclusions, and potential significance of this study.

4.     Line 24-27, too many keywords, and too long

5.     Line 31-32, is the shift to more sustainable and healthy diets a global goal or does it vary from place to place?

6.     In the introduction, threats to food security such as climate change should be mentioned.

7.     Figure 1, incoherence exists in the figure and figure legend: “R&I” and “RI”.

8.     Line 126, the structure of the manuscript is not clear enough and a flowchart may help improve readability.

9.     Line 839, usually FAO suggest that "food security" includes the "quantity", "quality", and "safety" of food.

10.   Line 847-849, the conclusion is not very adequate and does not well summarize the findings, conclusions, implications, and the inspiration for future research.

Author Response

First of all, we would like to thank the Editor and the Reviewers for their interesting and constructive comments.

Required modifications are now added in all sections of the revised manuscript and highlighted in RED color.

Answers to the Reviewer 1 Comments

foods-2822232:Global food security and sustainability issues: the road to 2030 from nutrition and sustainable healthy diets to food systems change-a case study of plant-based proteins and their biological and chemical modification

Focusing on global food security issues, this study discussed 11 pathways from farm to fork, trying to find effective guarantees for global food security. The authors seem to have attempted to emphasize the importance of plant-based proteins, but the organization and presentation of the manuscript are not sufficient for this purpose.

Answer: We gratefully appreciate your time, patience, and helpful comments regarding our manuscript. According to your advice, the manuscript has been revised according to your comments, please find below our responses, pointed out one by one.

  • Some specific comments are as follows: 
  1. Line 1-5, The title of the article is not enough to convey the full content of the article and is even a bit biased.

Answer:  As suggested by the Reviewer, we have added revised the Title as follow:

Global food security and sustainability issues: the road to 2030 from nutrition and sustainable healthy diets to food systems change

Please check the revised version

  1. Line 7, institution, city and country are not written in a clear order

Answer:  Dear Reviewer, we made the required changes as follow

Department of Food Science and Technology, University of the Peloponnese, Antikalamos, 24100 Kalamata, Greece University of the Peloponnese, [email protected]

Please check the revised version

  1. Line 12-23, the content of the abstract is not centered on the theme or title of the manuscript and does not address the findings, conclusions, and potential significance of this study.

Answer:  Dear reviewer, you are quite right, according to your remark; we have corrected the paper as suggested

We added this sentence

Accomplishing food/nutrition security for all across sustainable food systems (SFS) is governing by the Sustainable Development Goals (SDGs). SFS was connected with the entire SDGs viz. traditional framework of social inclusion, economic development, and environmental safety, inclusivity, and corporation extents of sustainable food systems. We debate that for the world to achieve sustainable development, a shift to SFS is necessary to guarantee food/nutrition security for all while operating within planetary boundaries to protect ecosystems and adapt to and mitigate climate change. Therefore, there is a requirement for original approaches that implement systemic and more participatory methods to engage with a wider range of food systems stakeholders. However, the deficiency of skills and tools regarding novel methodologies for food systems transformation is a key interference to the deployment of such approaches in practice. In a first part of this review, a summary of some challenges that occur in the governance of food system transformation was discussed. Through a case study plant-based proteins and their biological and chemical modification as dietary shift towards alternative proteins, we demonstrated that resource-efficient food systems and food waste on system transformation is useful in understanding both how (i) food system transformation has ensued, and (ii) how required transformations is prohibited.  Finally, we discussed the implications for governing food system transformations in terms of the nutrition and sustainable healthy diets that are needed to make systems change on food safety systems of the future. The linkage of food and the environment is evident focusing on nutrition and sustainable healthy diets. This would have not been accomplished without systems change and research towards new foods and more specifically new proteins such as plant-based and their biological and chemical modification.

Please check the revised version

  1. Line 24-27, too many keywords, and too long

Answer:  Dear Reviewer, we made the required changes as follow

Keywords:

Governance; Food safety; Food systems transformation; Sustainability transitions

Please check the revised version

  1. Line 31-32, is the shift to more sustainable and healthy diets a global goal or does it vary from place to place?

Answer:  Dear Editor, thank you for your comment, this part has been revised

We added this sentence

Owing to the pressures provoked by present, allied, global food systems to health/environmental degeneracy, challenges to renovate them in a more sustainable way are progressively emerging all-across the world. Above all, the tendency is to change from individualized agendas to cooperative strategies that can successfully promote the authentic transformation of food systems to be more maintainable. In this sense, according to European Commission (2020) [1] food system transformation is required in order to shift towards a more sustainable and healthy diet ensuring holistic food and nutrition security.

Please check the revised version

  1. In the introduction, threats to food security such as climate change should be mentioned.

Answer:  Dear Editor, thank you for your comment, this part has been revised

We added this sub section

Impact of climate change on food security

The biodiversity is an essential source of food. In this line, distresses over species disappearance are necessary by reason of services produced like pollination, pest, food and medicine controls. As an illustration, between: 1996-2003, the precipitations in parts of equatorial, East Africa, provoked flooding, reduction of crops, and agricultural yields [6]. Consequently, change in climatic has a direct impact on food production and distribution [7]. Firstly, an increase of incidence of pests and diseases was observed, and a loss of biodiversity, a decline of ecosystem functioning was noted. Secondly, the accessibility of water for crops, and fish production, and sea-level rise was reduced [8]. These sufferers include loss of life and food security status of millions of people in disaster-prone areas. Through extreme weather, CC will disturb food security and the crop yields too. By 2050, it is projected that agricultural yields in Africa alone could weakening by >30 % [9].

On the other hand, food preparation, processing, acquisition, distribution and consumption are impacted by CC [10] which influenced the plant and animal growth, water cycles, biodiversity and nutrient cycling, and the ways in which those are managed for agricultural practices, and food production [11]. In addition, CC could amend suitable cultivation zones with a wide range of crops.

CC influence on income-earning balances which can touch the ability to buy food, and a changing climate or climate extremes may affect the availability of certain food products. For example, in Tunisia and Egypt, there have been augmented prices of basic foodstuffs [12].

CC augmented the genetic erosion of landraces and threatening wild species, including crop wild relatives [13]. As a result, the existing varieties could be lost as farmers are replacing them with other landraces and improved varieties that are better adapted to the new conditions.

Added references

  1. Funk, W. C., Blouin, M. S., Corn, P. S., Maxell, B. A., Pilliod, D. S., Amish, S., & Allendorf, F. W. (2005). Population structure of Columbia spotted frogs (Rana luteiventris) is strongly affected by the landscape. Molecular ecology, 14(2), 483-496.
  2. Meybeck, A., Laval, E., Lévesque, R., & Parent, G. (2017, September). Food security and nutrition in the age of climate change. In Proceedings of the International Symposium organized by the Government of Québec in collaboration with FAO. Québec City (p. 132).
  3. IPCC, I. P. O. C. C. (2002). Climate change and biodiversity. IPCC Technical paper V. Group, 24, 77.
  4. Juma, S. G., & Kelonye, F. (2016). Projected rainfall and temperature changes over Bungoma county in western Kenya by the year 2050 based precis modeling system. Ethiopian Journal of Environmental Studies and Management, 9(5), 625-640.
  5. 10. Wheeler, T., & Von Braun, J. (2013). Climate change impacts on global food security. Science, 341(6145), 508-513.
  6. Yadav, S. S., Redden, R. J., Hatfield, J. L., Lotze-Campen, H., & Hall, A. E. (2011). Crop adaptation to climate change. John Wiley & Sons.
  7. Sasson, A. (2012). Food security for Africa: an urgent global challenge. Agriculture & Food Security, 1(1), 1-16.
  8. Jarvis, A., Lane, A., & Hijmans, R. J. (2008). The effect of climate change on crop wild relatives. Agriculture, Ecosystems & Environment, 126(1-2), 13-23.

Please check the revised version

  1. Figure 1, incoherence exists in the figure and figure legend: “R&I” and “RI”.

Answer: Dear reviewer, you are quite right, according to your remark; we have corrected the mistake as suggested

Fig. 1 Research & Innovation for future-proofing food systems (Adapted from European Commission, 2020) and figure modified.

Please check the revised version

  1. Line 126, the structure of the manuscript is not clear enough and a flowchart may help improve readability.

Answer: Dear reviewer, you are quite right, according to your remark; we have added this figure in the revised MS

Figure 2. Flowchart underlying governance, food systems transformation and sustainability transitions

Please check the revised version

  1. Line 839, usually FAO suggest that "food security" includes the "quantity", "quality", and "safety" of food.
  2. Line 847-849, the conclusion is not very adequate and does not well summarize the findings, conclusions, implications, and the inspiration for future research.

Answer:  Dear reviewer, according to your comment; we have corrected the conclusion as required.

Conclusions

The importance of science–policy–society interfaces (SPSIs) is outlined by transformation of food systems by connecting science, policy and society. On the other hand, Research and Innovation might be a strategic driver in the transformation near more sustainable food systems and thus characterizes a key planned area in the Farm to Fork Strategy and EU Green Deal. In this way, all stakeholders should not only be involved in discussing the interpretations of evidence but also in the policy process determining the action pathways to pursue towards quality, security and safety. In addition, capitalizing and underwriting to the proficiency development and tools for Research and Innovation food systems stakeholders to ‘do’ food system transformation is vital for supporting this transformation.

Answering to this gap, this current review has provided tools to lift capabilities between food systems stakeholders over the understanding both how food system transformation has ensued, and how required transformations is banned. Also, a deep discussion was reported on the implications for governing food system transformations in terms of the nutrition and sustainable healthy diets. These latter needed to make systems change on food safety systems of the future. The linkage of food and the environment is evident focusing on nutrition and a sustainable healthy diet was well established..

Governance and systems change, food security and safety, systems transformation and sustainability transitions have been well analyzed and underlined. All have their significance and the end result is how the consumers will interact with these systems and how their lives will be changed toward a more nutritional diet. Of course this necessitates that the environment will not be further affected negatively. In this direction a better dietary shift towards alternative proteins and more specifically plant-based proteins is imperative. Hence, the technical aspect of their biological and chemical modification is critically analyzed. This paper appeals all readers whether they work with food-, health- or in the food economy systems to apply, familiarize and integrate the Tools for Transformation to adopt more inclusive, transdisciplinary, and systemic approaches to the demanding challenges we face today.

Please check the revised version

Reviewer 2 Report

Comments and Suggestions for Authors

I think it would be worth clearly and unambiguously formulating the purpose of the undertaken review of the state of knowledge related to the approach to issues of safety and sustainability in the field of global food. It is true that the authors wrote in the Abstract what they tried to do, but the same could be written in the form of the aim of a review study, including the justification for dealing with the topic under consideration. This justification may result from a gap in the current state of knowledge in the area of food, and filling this gap required a review. Therefore, it would be worth identifying this gap and formulating it in the initial part of the article.

In my opinion, it would be worth limiting the number of proposed keywords in the article. The proposed keywords are very long phrases; I know that they result from the specificity of the discussed topic, but it is worth considering shortening them, and perhaps including acronyms would be a suggestion; a large number of such acronyms are given in the article. The long phrases included in the set of keywords are justified by the considered ideas in the food economy system, but perhaps the authors could find a compromise when formulating the keywords, both in terms of quantity and quality.

If the authors use acronyms in the article, e.g. CLIC (lines: 248, 249), it would be worth providing their full name (full expansion). There is no such expansion of the acronym even in the cited publication (line: 969), so it is unclear what it is about. The same remark applies to the acronym SUSFANS, which was mentioned several times in the article, for the first time in line: 141. The first time an acronym is mentioned in the text, it is worth providing its full name.

Since a lot of acronyms were used in the article, in my opinion it is worth considering listing all the acronyms in one place, for example directly in the article, but even better in a file opened using the link provided in the article (such a file can be placed on a platform available via the link) . This would make it much easier for the reader to use these acronyms and identify them clearly in terms of relating the acronyms to their full names.

If the authors referred to the idea of "from farm to fork" in the article, in my opinion it would also be worth developing basic knowledge about the idea of "from fork to farm". In this way, the article would present a more complete picture of the approach to sustainable food production, including specific examples from practice.

Perhaps it would be worth mentioning in the article other ideas related to the approach to the production of selected groups of products, e.g. milk and dairy production. An example of such an idea is "from grass to glass". With this idea, the issue of limitations in introducing various solutions (ideas) in the food economy system and the chain connecting the place of obtaining agricultural raw materials and the place of their consumption could be developed.

In some places in the article it would be worth tidying up the citation rules. For example, in lines 275-276 the publication [49] is quoted at the end of the sentence and at the same time the same publication is quoted in brackets (). I think there is no need to cite the same publication twice in the same place. A similar approach, i.e. double quoting literature sources, was used by the authors, for example, in the paragraph on lines: 814-824.

Generally, the article is very long and it might be worth considering shortening it a bit. The article takes into account many ideas/approaches to assessing the safety and sustainability of food systems. This may make it difficult for the reader to analyze the article and isolate key aspects of the assessment of food management systems.

In my opinion, the article lacks a slightly deeper discussion of the authors' own (the authors' own interpretation) summarizing the considered actions regarding food and food systems. The conclusions formulated are very general and basically constitute a repetition of selected information in the basic part of the article. In my opinion, the Conclusions could be written in such a way as to take into account the Authors' own reflections resulting from the review. In the Conclusions, it is also worth trying to outline the future of the development of the food system.

Author Response

Answers to the Reviewer 2 Comments

I think it would be worth clearly and unambiguously formulating the purpose of the undertaken review of the state of knowledge related to the approach to issues of safety and sustainability in the field of global food.

We gratefully appreciate your time, patience, and helpful comments regarding our manuscript. According to your advice, the manuscript has been revised according to your comments, please find below our responses, pointed out one by one.

It is true that the authors wrote in the Abstract what they tried to do, but the same could be written in the form of the aim of a review study, including the justification for dealing with the topic under consideration. This justification may result from a gap in the current state of knowledge in the area of food, and filling this gap required a review. Therefore, it would be worth identifying this gap and formulating it in the initial part of the article.

Answer:  Dear reviewer, you are quite right, according to your remark; we have corrected the paper as suggested

We added these sentences:

Abstract section

Accomplishing food/nutrition security for all across sustainable food systems (SFS) is governing by the Sustainable Development Goals (SDGs). SFS was connected with the entire SDGs viz. traditional framework of social inclusion, economic development, and environmental safety, inclusivity, and corporation extents of sustainable food systems. We debate that for the world to achieve sustainable development, a shift to SFS is necessary to guarantee food/nutrition security for all while operating within planetary boundaries to protect ecosystems and adapt to and mitigate climate change. Therefore, there is a requirement for original approaches that implement systemic and more participatory methods to engage with a wider range of food systems stakeholders. However, the deficiency of skills and tools regarding novel methodologies for food systems transformation is a key interference to the deployment of such approaches in practice. In a first part of this review, a summary of some challenges that occur in the governance of food system transformation was discussed. Through a case study plant-based proteins and their biological and chemical modification as dietary shift towards alternative proteins, we demonstrated that resource-efficient food systems and food waste on system transformation is useful in understanding both how (i) food system transformation has ensued, and (ii) how required transformations is prohibited.  Finally, we discussed the implications for governing food system transformations in terms of the nutrition and sustainable healthy diets that are needed to make systems change on food safety systems of the future. The linkage of food and the environment is evident focusing on nutrition and sustainable healthy diets. This would have not been accomplished without systems change and research towards new foods and more specifically new proteins such as plant-based and their biological and chemical modification.

At the end of the introduction section

In this review, a summary of some challenges that occur in the governance of food system transformation was firstly investigated. Through a case study plant-based proteins and their biological and chemical modification as dietary shift towards alternative proteins, we demonstrated that resource-efficient food systems and food waste on system transformation is useful in understanding both how (i) food system transformation has ensued, and (ii) how required transformations is prohibited.  Finally, we discussed the implications for governing food system transformations in terms of the nutrition and sustainable healthy diets that are needed to make systems change on food safety systems of the future (Fig. 2).

Figure 2. Flowchart underlying governance, food systems transformation and sustainability transitions

Please check the revised version

In my opinion, it would be worth limiting the number of proposed keywords in the article. The proposed keywords are very long phrases; I know that they result from the specificity of the discussed topic, but it is worth considering shortening them, and perhaps including acronyms would be a suggestion; a large number of such acronyms are given in the article. The long phrases included in the set of keywords are justified by the considered ideas in the food economy system, but perhaps the authors could find a compromise when formulating the keywords, both in terms of quantity and quality.

Answer:  Dear Reviewer, we made the required changes as follow

Keywords:

Governance; Food safety; Food systems transformation; Sustainability transitions

Please check the revised version

  1. If the authors use acronyms in the article, e.g. CLIC (lines: 248, 249), it would be worth providing their full name (full expansion). There is no such expansion of the acronym even in the cited publication (line: 969), so it is unclear what it is about. The same remark applies to the acronym SUSFANS, which was mentioned several times in the article, for the first time in line: 141. The first time an acronym is mentioned in the text, it is worth providing its full name.

Since a lot of acronyms were used in the article, in my opinion it is worth considering listing all the acronyms in one place, for example directly in the article, but even better in a file opened using the link provided in the article (such a file can be placed on a platform available via the link) . This would make it much easier for the reader to use these acronyms and identify them clearly in terms of relating the acronyms to their full names.

Answer:  Dear Reviewer, you are quite right, according to your remark; we have corrected the paper as suggested

Please check the revised version

If the authors referred to the idea of "from farm to fork" in the article, in my opinion it would also be worth developing basic knowledge about the idea of "from fork to farm". In this way, the article would present a more complete picture of the approach to sustainable food production, including specific examples from practice.

Answer:  Dear reviewer, you are quite right, according to your remark; we have corrected the paper as suggested

We added these sentences:

  1. Food safety systems of the future

To turn into a climate-neutral continent by 2050, the EU has propelled the Green Deal, with the Farm to Fork and the Biodiversity Strategy as its foundations [181]. The goal of the Farm to Fork Strategy is to evolution to a maintainable food system [182]. Biodiversity Strategy involved of numerous proposed policy actions with the goal of paying to lower greenhouse gas emissions, biodiversity conservation, condensed pesticide use, and augmented consumer empowerment [183]. The Farm to Fork Strategy predicts creation the food system more justifiable by striking restrictions on fertilizer and pesticides and casing at least 25% of the EU’s agricultural land under organic farming by 2030  [182]. Farm to Fork Strategy and the Biodiversity Strategy targeted to elnance European biodiversity levels by growing the agricultural land amount under high-diversity landscape features to at least 10% [182]. However, there are worries regarding whether organic production and certification are adequate to preserve and expand biodiversity and to spread climate targets as defined in the UN Sustainable Development Goals. Therefore, holistic approaches that go beyond organic production which also address the role of consumers in food labelling are suggested [184]. In this line, the ecological food system framework initiative was hurled with the meaning of making the EU food system sustainable and integrating sustainability into all food-related policies [182]. While progress has been made in quantifying product-specific environmental impacts, major challenges in how to track and communicate these influences continue. For instance, there is a lack of EU-wide monitoring frameworks for appraising progress towards food sustainability objectives. The initiative on green claims was launched to substantiate green product credentials against a regular methodology to evaluate their impact on the environment.

Added references

Bazzan, G., Daugbjerg, C., & Tosun, J. (2023). Attaining policy integration through the integration of new policy instruments: The case of the Farm to Fork Strategy. Applied Economic Perspectives and Policy45(2), 803-818.

Schulze, C., Matzdorf, B., Rommel, J., Czajkowski, M., García-Llorente, M., Gutiérrez-Briceño, I., ... & Zawadzki, W. (2024). Between farms and forks: Food industry perspectives on the future of EU food labelling. Ecological Economics217, 108066.

Schebesta, H., & Candel, J. J. (2020). Game-changing potential of the EU’s Farm to Fork Strategy. Nature Food1(10), 586-588.

Riccaboni, A., Neri, E., Trovarelli, F., & Pulselli, R. M. (2021). Sustainability-oriented research and innovation in ‘farm to fork’value chains. Current Opinion in Food Science42, 102-112

Please check the revised version

Perhaps it would be worth mentioning in the article other ideas related to the approach to the production of selected groups of products, e.g. milk and dairy production. An example of such an idea is "from grass to glass". With this idea, the issue of limitations in introducing various solutions (ideas) in the food economy system and the chain connecting the place of obtaining agricultural raw materials and the place of their consumption could be developed.

Answer:  Dear reviewer, you are quite right, according to your remark; we have corrected the paper as suggested

We added these sentences:

As an illustration regarding milk and dairy production in the food economy system and the chain connecting: the attitudes of a group of young Polish consumers towards selected features of dairy products [172]. Consumer opinion surveys include key elements of improving the food market and assessing consumers’ approaches to current issues related to access to high-quality food. The Gaworski et al. [172] investigation aimed to find out the attitudes of young Polish consumers about dairy products. The objective was linked to the valuation of selected features of dairy products and their packaging, assessment of regional products and novelties in dairy production. These consumers pointed on the importance of quality/taste of dairy products, and designated the small role of packaging. Additionally, major of respondents designated that they did not notice to the biodegradability packaging. When asked about regional dairy products, respondents paid great attention to their value resultant from natural methods of production without preservatives. However, a small % of young respondents showed knowledge of the idea of dairy production ‘from grass to glass’, which would indicate inadequate interest in innovative solutions in the dairy sector.

Added reference

  1. Gaworski, M., Borowski, P. F., & Zajkowska, M. (2021). Attitudes of a group of young Polish consumers towards selected features of dairy products.

Please check the revised version

In some places in the article it would be worth tidying up the citation rules. For example, in lines 275-276 the publication [49] is quoted at the end of the sentence and at the same time the same publication is quoted in brackets (). I think there is no need to cite the same publication twice in the same place. A similar approach, i.e. double quoting literature sources, was used by the authors, for example, in the paragraph on lines: 814-824.

Answer:  Dear Reviewer, you are quite right, according to your remark; we have corrected the paper as suggested

Please check the revised version

Generally, the article is very long and it might be worth considering shortening it a bit. The article takes into account many ideas/approaches to assessing the safety and sustainability of food systems. This may make it difficult for the reader to analyze the article and isolate key aspects of the assessment of food management systems.

Answer: Dear reviewer, the whole MS was revised, and some parts and sentences was deleted. In addition, all parts were linked

In my opinion, the article lacks a slightly deeper discussion of the authors' own (the authors' own interpretation) summarizing the considered actions regarding food and food systems. The conclusions formulated are very general and basically constitute a repetition of selected information in the basic part of the article. In my opinion, the Conclusions could be written in such a way as to take into account the Authors' own reflections resulting from the review.

Answer: Dear reviewer, the whole MS was revised, and some parts and sentences was deleted. In addition, All parts were linked

Please check the revised version

  1. In the Conclusions, it is also worth trying to outline the future of the development of the food system.

Answer:  Dear reviewer, according to your comment; we have corrected the conclusion as required.

Conclusions

The importance of science–policy–society interfaces (SPSIs) is outlined by transformation of food systems by connecting science, policy and society. On the other hand, Research and Innovation might be a strategic driver in the transformation near more sustainable food systems and thus characterizes a key planned area in the Farm to Fork Strategy and EU Green Deal. In this way, all stakeholders should not only be involved in discussing the interpretations of evidence but also in the policy process determining the action pathways to pursue towards quality, security and safety. In addition, capitalizing and underwriting to the proficiency development and tools for Research and Innovation food systems stakeholders to ‘do’ food system transformation is vital for supporting this transformation.

Answering to this gap, this current review has provided tools to lift capabilities between food systems stakeholders over the understanding both how food system transformation has ensued, and how required transformations is banned. Also, a deep discussion was reported on the implications for governing food system transformations in terms of the nutrition and sustainable healthy diets. These latter needed to make systems change on food safety systems of the future. The linkage of food and the environment is evident focusing on nutrition and a sustainable healthy diet was well established..

Governance and systems change, food security and safety, systems transformation and sustainability transitions have been well analyzed and underlined. All have their significance and the end result is how the consumers will interact with these systems and how their lives will be changed toward a more nutritional diet. Of course this necessitates that the environment will not be further affected negatively. In this direction a better dietary shift towards alternative proteins and more specifically plant-based proteins is imperative. Hence, the technical aspect of their biological and chemical modification is critically analyzed. This paper appeals all readers whether they work with food-, health- or in the food economy systems to apply, familiarize and integrate the Tools for Transformation to adopt more inclusive, transdisciplinary, and systemic approaches to the demanding challenges we face today.

Please check the revised version

Reviewer 3 Report

Comments and Suggestions for Authors

The review provides a comprehensive overview of the environmental and ecological issues related to sustainable food systems. It highlights the importance of adopting a farm-to-fork approach that involves all stakeholders and considers the entire food supply chain. The inclusion of eleven pathways for action is a valuable aspect of the review, as it offers a roadmap for addressing the challenges faced by food systems. Also, the review covers a broad range of important topics related to global food security and sustainability in a comprehensive manner. It discusses key issues like governance, urban food systems, alternative proteins, nutrition, food safety, etc. in depth, citing relevant literature. The scope and content seem scientifically valid. 

One strength of the review is its emphasis on the linkage between food and the environment. The authors correctly recognize that sustainable food systems must prioritize nutrition and healthy diets, while also considering the impact of food production on the environment. The focus on new foods and proteins, particularly plant-based options, aligns with the growing interest in alternative protein sources as a means of reducing the environmental footprint of food systems. 

However, the review could benefit from a more concise and structured presentation. The inclusion of eleven pathways for action, while comprehensive, may overwhelm readers and make it difficult to grasp the key points. Additionally, the review could provide more specific examples or case studies to illustrate the challenges and potential solutions discussed.

In summary, based on the points outlined above, the review article seems scientifically valid, technically accurate, reproducible, and ethically sound.

Overall, the review provides a solid foundation for a research paper on environmental and ecological issues in sustainable food systems. With some refinement and additional detail, it could effectively engage readers and encourage further exploration of this important topic.

Author Response

Answers to the Reviewer 3 Comments

The review provides a comprehensive overview of the environmental and ecological issues related to sustainable food systems. It highlights the importance of adopting a farm-to-fork approach that involves all stakeholders and considers the entire food supply chain. The inclusion of eleven pathways for action is a valuable aspect of the review, as it offers a roadmap for addressing the challenges faced by food systems. Also, the review covers a broad range of important topics related to global food security and sustainability in a comprehensive manner. It discusses key issues like governance, urban food systems, alternative proteins, nutrition, food safety, etc. in depth, citing relevant literature. The scope and content seem scientifically valid.

One strength of the review is its emphasis on the linkage between food and the environment. The authors correctly recognize that sustainable food systems must prioritize nutrition and healthy diets, while also considering the impact of food production on the environment. The focus on new foods and proteins, particularly plant-based options, aligns with the growing interest in alternative protein sources as a means of reducing the environmental footprint of food systems.

We gratefully appreciate your time, patience, and helpful comments regarding our manuscript. According to your advice, the manuscript has been revised according to your comments, please find below our responses, pointed out one by one.

However, the review could benefit from a more concise and structured presentation. The inclusion of eleven pathways for action, while comprehensive, may overwhelm readers and make it difficult to grasp the key points. Additionally, the review could provide more specific examples or case studies to illustrate the challenges and potential solutions discussed.

In summary, based on the points outlined above, the review article seems scientifically valid, technically accurate, reproducible, and ethically sound.

Overall, the review provides a solid foundation for a research paper on environmental and ecological issues in sustainable food systems. With some refinement and additional detail, it could effectively engage readers and encourage further exploration of this important topic.

Answer:  Dear reviewer, thank you for your comment, all MS part has been revised as recommended. The whole MS was revised, and some parts and sentences was deleted. In addition, all parts were linked

We added these sentences:

  1. Abstract section

Accomplishing food/nutrition security for all across sustainable food systems (SFS) is governing by the Sustainable Development Goals (SDGs). SFS was connected with the entire SDGs viz. traditional framework of social inclusion, economic development, and environmental safety, inclusivity, and corporation extents of sustainable food systems. We debate that for the world to achieve sustainable development, a shift to SFS is necessary to guarantee food/nutrition security for all while operating within planetary boundaries to protect ecosystems and adapt to and mitigate climate change. Therefore, there is a requirement for original approaches that implement systemic and more participatory methods to engage with a wider range of food systems stakeholders. However, the deficiency of skills and tools regarding novel methodologies for food systems transformation is a key interference to the deployment of such approaches in practice. In a first part of this review, a summary of some challenges that occur in the governance of food system transformation was discussed. Through a case study plant-based proteins and their biological and chemical modification as dietary shift towards alternative proteins, we demonstrated that resource-efficient food systems and food waste on system transformation is useful in understanding both how (i) food system transformation has ensued, and (ii) how required transformations is prohibited.  Finally, we discussed the implications for governing food system transformations in terms of the nutrition and sustainable healthy diets that are needed to make systems change on food safety systems of the future. The linkage of food and the environment is evident focusing on nutrition and sustainable healthy diets. This would have not been accomplished without systems change and research towards new foods and more specifically new proteins such as plant-based and their biological and chemical modification.

  1. At the end of the introduction section

In this review, a summary of some challenges that occur in the governance of food system transformation was firstly investigated. Through a case study plant-based proteins and their biological and chemical modification as dietary shift towards alternative proteins, we demonstrated that resource-efficient food systems and food waste on system transformation is useful in understanding both how (i) food system transformation has ensued, and (ii) how required transformations is prohibited.  Finally, we discussed the implications for governing food system transformations in terms of the nutrition and sustainable healthy diets that are needed to make systems change on food safety systems of the future (Fig. 2).

Figure 2. Flowchart underlying governance, food systems transformation and sustainability transitions

  1. Impact of climate change on food security

The biodiversity is an essential source of food. In this line, distresses over species disappearance are necessary by reason of services produced like pollination, pest, food and medicine controls. As an illustration, between: 1996-2003, the precipitations in parts of equatorial, East Africa, provoked flooding, reduction of crops, and agricultural yields [6]. Consequently, change in climatic has a direct impact on food production and distribution [7]. Firstly, an increase of incidence of pests and diseases was observed, and a loss of biodiversity, a decline of ecosystem functioning was noted. Secondly, the accessibility of water for crops, and fish production, and sea-level rise was reduced [8]. These sufferers include loss of life and food security status of millions of people in disaster-prone areas. Through extreme weather, CC will disturb food security and the crop yields too. By 2050, it is projected that agricultural yields in Africa alone could weakening by >30 % [9].

On the other hand, food preparation, processing, acquisition, distribution and consumption are impacted by CC [10] which influenced the plant and animal growth, water cycles, biodiversity and nutrient cycling, and the ways in which those are managed for agricultural practices, and food production [11]. In addition, CC could amend suitable cultivation zones with a wide range of crops.

CC influence on income-earning balances which can touch the ability to buy food, and a changing climate or climate extremes may affect the availability of certain food products. For example, in Tunisia and Egypt, there have been augmented prices of basic foodstuffs [12].

CC augmented the genetic erosion of landraces and threatening wild species, including crop wild relatives [13]. As a result, the existing varieties could be lost as farmers are replacing them with other landraces and improved varieties that are better adapted to the new conditions.

Added references

  1. Funk, W. C., Blouin, M. S., Corn, P. S., Maxell, B. A., Pilliod, D. S., Amish, S., & Allendorf, F. W. (2005). Population structure of Columbia spotted frogs (Rana luteiventris) is strongly affected by the landscape. Molecular ecology, 14(2), 483-496.
  2. Meybeck, A., Laval, E., Lévesque, R., & Parent, G. (2017, September). Food security and nutrition in the age of climate change. In Proceedings of the International Symposium organized by the Government of Québec in collaboration with FAO. Québec City (p. 132).
  3. IPCC, I. P. O. C. C. (2002). Climate change and biodiversity. IPCC Technical paper V. Group, 24, 77.
  4. Juma, S. G., & Kelonye, F. (2016). Projected rainfall and temperature changes over Bungoma county in western Kenya by the year 2050 based precis modeling system. Ethiopian Journal of Environmental Studies and Management, 9(5), 625-640.
  5. 10. Wheeler, T., & Von Braun, J. (2013). Climate change impacts on global food security. Science, 341(6145), 508-513.
  6. Yadav, S. S., Redden, R. J., Hatfield, J. L., Lotze-Campen, H., & Hall, A. E. (2011). Crop adaptation to climate change. John Wiley & Sons.
  7. Sasson, A. (2012). Food security for Africa: an urgent global challenge. Agriculture & Food Security, 1(1), 1-16.
  8. Jarvis, A., Lane, A., & Hijmans, R. J. (2008). The effect of climate change on crop wild relatives. Agriculture, Ecosystems & Environment, 126(1-2), 13-23.

We added these sentences:

  1. Food safety systems of the future

To turn into a climate-neutral continent by 2050, the EU has propelled the Green Deal, with the Farm to Fork and the Biodiversity Strategy as its foundations [181]. The goal of the Farm to Fork Strategy is to evolution to a maintainable food system [182]. Biodiversity Strategy involved of numerous proposed policy actions with the goal of paying to lower greenhouse gas emissions, biodiversity conservation, condensed pesticide use, and augmented consumer empowerment [183]. The Farm to Fork Strategy predicts creation the food system more justifiable by striking restrictions on fertilizer and pesticides and casing at least 25% of the EU’s agricultural land under organic farming by 2030  [182]. Farm to Fork Strategy and the Biodiversity Strategy targeted to elnance European biodiversity levels by growing the agricultural land amount under high-diversity landscape features to at least 10% [182]. However, there are worries regarding whether organic production and certification are adequate to preserve and expand biodiversity and to spread climate targets as defined in the UN Sustainable Development Goals. Therefore, holistic approaches that go beyond organic production which also address the role of consumers in food labelling are suggested [184]. In this line, the ecological food system framework initiative was hurled with the meaning of making the EU food system sustainable and integrating sustainability into all food-related policies [182]. While progress has been made in quantifying product-specific environmental impacts, major challenges in how to track and communicate these influences continue. For instance, there is a lack of EU-wide monitoring frameworks for appraising progress towards food sustainability objectives. The initiative on green claims was launched to substantiate green product credentials against a regular methodology to evaluate their impact on the environment.

Added references

  1. Bazzan, G., Daugbjerg, C., & Tosun, J. (2023). Attaining policy integration through the integration of new policy instruments: The case of the Farm to Fork Strategy. Applied Economic Perspectives and Policy45(2), 803-818.
  2. Schulze, C., Matzdorf, B., Rommel, J., Czajkowski, M., García-Llorente, M., Gutiérrez-Briceño, I., ... & Zawadzki, W. (2024). Between farms and forks: Food industry perspectives on the future of EU food labelling. Ecological Economics217, 108066.
  3. Schebesta, H., & Candel, J. J. (2020). Game-changing potential of the EU’s Farm to Fork Strategy. Nature Food1(10), 586-588.
  4. Riccaboni, A., Neri, E., Trovarelli, F., & Pulselli, R. M. (2021). Sustainability-oriented research and innovation in ‘farm to fork’value chains. Current Opinion in Food Science42, 102-112

5. Conclusions

The importance of science–policy–society interfaces (SPSIs) is outlined by transformation of food systems by connecting science, policy and society. On the other hand, Research and Innovation might be a strategic driver in the transformation near more sustainable food systems and thus characterizes a key planned area in the Farm to Fork Strategy and EU Green Deal. In this way, all stakeholders should not only be involved in discussing the interpretations of evidence but also in the policy process determining the action pathways to pursue towards quality, security and safety. In addition, capitalizing and underwriting to the proficiency development and tools for Research and Innovation food systems stakeholders to ‘do’ food system transformation is vital for supporting this transformation.

µAnswering to this gap, this current review has provided tools to lift capabilities between food systems stakeholders over the understanding both how food system transformation has ensued, and how required transformations is banned. Also, a deep discussion was reported on the implications for governing food system transformations in terms of the nutrition and sustainable healthy diets. These latter needed to make systems change on food safety systems of the future. The linkage of food and the environment is evident focusing on nutrition and a sustainable healthy diet was well established..

Governance and systems change, food security and safety, systems transformation and sustainability transitions have been well analyzed and underlined. All have their significance and the end result is how the consumers will interact with these systems and how their lives will be changed toward a more nutritional diet. Of course this necessitates that the environment will not be further affected negatively. In this direction a better dietary shift towards alternative proteins and more specifically plant-based proteins is imperative. Hence, the technical aspect of their biological and chemical modification is critically analyzed. This paper appeals all readers whether they work with food-, health- or in the food economy systems to apply, familiarize and integrate the Tools for Transformation to adopt more inclusive, transdisciplinary, and systemic approaches to the demanding challenges we face today.

Please check the revised version